# CAD-ALZ: A Blockwise Fine-Tuning Strategy on Convolutional Model and Random Forest Classifier for Recognition of Multistage Alzheimer’s Disease

**DOI:** 10.3390/diagnostics13010167

**Published:** 2023-01-03

**Authors:** Qaisar Abbas, Ayyaz Hussain, Abdul Rauf Baig

**Affiliations:** 1College of Computer and Information Sciences, Imam Mohammad Ibn Saud Islamic University (IMSIU), Riyadh 11432, Saudi Arabia; 2Department of Computer Science, Quaid-i-Azam University, Islamabad 44000, Pakistan

**Keywords:** Alzheimer’s, inception-v3, magnetic resonance imaging, mental deterioration, random forest, separable CNN, transfer learning

## Abstract

Mental deterioration or Alzheimer’s (ALZ) disease is progressive and causes both physical and mental dependency. There is a need for a computer-aided diagnosis (CAD) system that can help doctors make an immediate decision. (1) Background: Currently, CAD systems are developed based on hand-crafted features, machine learning (ML), and deep learning (DL) techniques. Those CAD systems frequently require domain-expert knowledge and massive datasets to extract deep features or model training, which causes problems with class imbalance and overfitting. Additionally, there are still manual approaches used by radiologists due to the lack of dataset availability and to train the model with cost-effective computation. Existing works rely on performance improvement by neglecting the problems of the limited dataset, high computational complexity, and unavailability of lightweight and efficient feature descriptors. (2) Methods: To address these issues, a new approach, CAD-ALZ, is developed by extracting deep features through a ConvMixer layer with a blockwise fine-tuning strategy on a very small original dataset. At first, we apply the data augmentation method to images to increase the size of datasets. In this study, a blockwise fine-tuning strategy is employed on the ConvMixer model to detect robust features. Afterwards, a random forest (RF) is used to classify ALZ disease stages. (3) Results: The proposed CAD-ALZ model obtained significant results by using six evaluation metrics such as the F1-score, Kappa, accuracy, precision, sensitivity, and specificity. The CAD-ALZ model performed with a sensitivity of 99.69% and an F1-score of 99.61%. (4) Conclusions: The suggested CAD-ALZ approach is a potential technique for clinical use and computational efficiency compared to state-of-the-art approaches. The CAD-ALZ model code is freely available on GitHub for the scientific community.

## 1. Introduction

Mental deterioration (MD) or Alzheimer’s (ALZ) disease is the most prevalent neurodegenerative [1] illness among elderly people, and it causes 60% to 80% of cases of dementia. According to one study, there are around 5 million ALZ sufferers in the US alone, and if the prevalence rate does not change, that number could triple by 2050. It is characterized by cognitive impairment and behavioral disorders. The early detection of ALZ is crucial because there are currently no medications that can halt or reverse the disease’s progression [2]. Such an early diagnosis gives the patient knowledge of the severity and enables them to take preventative actions, such as medication and lifestyle changes. Due to the range and dynamic character of the symptoms, clinical diagnosis is difficult [3]. Magnetic resonance imaging (MRI) is one of the non-invasive neuroimaging techniques which shows a wealth of information about the structural integrity of the brain tissues. Moreover, it offers good contrast between gray matter and white matter. As a result, it has frequently been employed to investigate the differences between the brains of ALZ patients and cognitively normal people. The subjectivity and workload associated with the extraction of handcrafted features, however, make these strategies ineffective. In the past, feature extraction and classification were the first two primary steps designed in a computer-aided diagnosis (CAD) system for automatic detection of ALZ.

Today, a variety of deep-learning (DL)-based techniques are being used to diagnose ALZ disease. These strategies include feeding the whole brains of 2D or 3D image patches or region-of-interests (ROIs) into a DL model [4]. As a result, putting the entire brain at once into a deep model typically yields an unreliable diagnosis, since irrelevant brain areas would undermine rather than enhance the model’s ability to distinguish between different conditions. It is a good idea to extract partially regular or randomly overlapped image patches to train a DL model. Furthermore, the brain anatomy is not considered by patch-based approaches. The extraction of ROIs based on a set of experimentally defined landmarks is extensively utilized as an alternative approach. Despite the enhanced performance, this technique is still heavily dependent on the accuracy of landmarks.

The transfer learning (TL) capabilities of DL models [5,6,7] were also used in past studies. To solve the problem of class imbalance, those studies introduced the first layers of the pre-trained AlexNet model to the input dataset. Three different models have been put out for the real-time and early diagnosis of ALZ. Initially, many classifiers were used to assess a manually generated feature extraction approach based on textural and statistical data. Afterwards, those studies established a pre-trained model, and it automatically identifies features from MRI images.

Numerous other classification-based strategies for the early diagnosis of ALZ disease have been presented in recent years. We have grouped the techniques after much analysis and thought. Binary and multi-class classification schemes are also included in the classification categories [8]. Alzheimer’s (ALZ) disease, often known as “MD” or “Alzheimer’s,” is a form of dementia characterized by physical and mental impairment that progresses through cognitive degeneration. This illness has multiple stages [9], starting with an individual who showed no signs of dementia and progressing to very mild, mild, and moderate dementia. Normal behavior and thinking processes are hampered, and everyday, ordinary tasks become dysfunctional. Four stages were identified from MRI images which were normal or with ALZ disease, including very mild, mild, and moderate dementia.

Despite the significant research efforts being made in this area, ALZ still has significant research gaps [9] because ALZ is a disease that progresses through many stages. Many Alzheimer’s patients can be spared a tragic outcome if the disease can be diagnosed in its early stages. In current neuroimaging-based research [10], ref. [11] indicates positive indicators for early and accurate identification of ALZ disease [12,13,14,15,16]. Although the terms “machine” and “deep learning” are relatively new, their methodologies have been applied to several medical problems, particularly in the domain of CAD. Many studies [17,18,19,20,21,22,23,24,25] have used deep learning (DL) and machine learning approaches to extract and categorize ALZ. A sizable number of labeled data samples are needed to train the network before it can be used to diagnose ALZ. Large, labeled data availability is still a problem.

In the field of medical images, the CNN trained with so many datasets is anticipated to yield extremely precise and effective outcomes [26,27,28,29]. A pre-trained network model with a large amount of data and only minor classification model adjustments is one of the many strategies that may be utilized with CNN to improve the recognition results in the domain of medicine. Since the conventional description methods with little feature aggregation have shown promising outcomes, a cutting-edge method for classifying brain images using DL was published by [30,31,32,33,34,35,36,37,38,39,40,41,42,43,44,45,46,47,48,49,50,51,52,53,54,55,56,57].

A real-time computer-aided diagnostic system that can process the input MRI pictures and categorize the patients as healthy or MD patients is urgently needed. For this, the system must be capable of accurately classifying patients’ MD phases in real-time, which can also be beneficial to doctors. A new approach with the addition of the ConvMixer [58] layer is used to blend excellent features from CNNs and patches based on a vision transformer on a very small original dataset. To solve issues, deep features are extracted using a ConvMixer architecture and then utilized in a random forest (RF) [59] to classify ALZ disease stages. The experiments are performed on the Kaggle-ALZ dataset [60]. We have also compared with latest the DL architecture [61] by using Kaggle-ALZ and ADNI [62] datasets. In this study, a blockwise fine-tuning method is used on the ConvMixer model to effectively detect and optimize DL-based features. In this study, we assess the performance of deep features for identifying the four stages of Alzheimer’s disease (ALZ) and suggest deep features based on a transfer learning (TL) strategy. The main contributions of this study are as follows:
(1)This study proposes the CAD-ALZ model for joint deep and robust feature extraction, which are critical to diagnose Alzheimer’s disease (ALZ).(2)This paper integrates a blockwise fine-tune strategy into ConvMixer to extract deep features. This strategy unit is proposed in place of the attention module that promotes the flow of information with more efficient computation.(3)The CAD-ALZ model offers a cutting-edge diagnosis for the identification of stages of ALZ disease in an efficient manner compared to other approaches.(4)Multiple classes of ALZ are greatly imbalanced. To balance and increase this dataset, a data augmentation technique is applied.(5)Extensive experiments are performed on two publicly available benchmarks such as Kaggle-ALZ and ADNI. A detailed comparative study was presented to compare the proposed strategy to other existing DL approaches.

This paper is organized as follows for the remaining portions. Section 2 discusses the materials and development methods used in this work. Section 3 presents the experimental results from the dataset. Section 4 presents the discussion, and Section 5 concludes this research.

## 2. Literature Review

Numerous classification-based strategies for the early diagnosis of mental deterioration or Alzheimer’s disease (ALZ) have been presented in recent years. Binary and multi-class classification schemes are also included in the classification categories. This ALZ had multiple stages, starting with an individual who showed no signs of dementia and progressing to very mild, mild, and moderate dementia. The latest studies utilized deep learning (DL) algorithms, which are briefly described in the subsequent sections and compared in Table 1.

Current studies indicate positive indicators for early and accurate identification of ALZ disease [12,13,14,15,16]. Many studies [17,18,19,20,21,22,23,24,25] have used deep learning (DL) and machine learning approaches to extract and categorize ALZ. A sizable number of labeled data samples are needed to train the network before it can be used to diagnose ALZ. Large-labeled data availability is still a challenging problem. In the field of medical images, the CNN trained with so many datasets is anticipated to yield extremely precise and effective outcomes [26,27,28,29]. Pre-trained network models [30,31,32,33,34,35,36,37,38,39,40,41,42,43,44,45,46,47,48,49,50,51,52,53,54,55,56,57] with a large amount of data and only minor classification model adjustments are one of the many strategies that may be utilized with CNN to improve the recognition results in the domain of medicine. The conventional description methods with little feature aggregation have shown promising outcomes. Most of these studies have been tested on one dataset, so those models have limited capability in terms of generalization. The cutting-edge DL methods, which are directly related to our studies, are described in the following paragraphs. 

A weakly supervised densely connected neural network (wiseDNN) [30] was developed to predict the prognosis of brain diseases by using MRI. They first extracted multiscale image patches from MRI. Afterwards, they created a weakly supervised wiseDNN network for task-oriented extraction of imaging features and joint prediction of multiple clinical measures. The experimental findings from the ADNI dataset were used to evaluate the network. The proposed four-stage classification method had an accuracy of 96.5%. However, they have been tested on a limited dataset, so the wiseDNN did not provided a generalized solution. In another study [31], the authors developed a different method based on a deformation-based technique and tested on the ADNI dataset to recognize two classes of ALZ disease. A new FCM-based Weighted Probabilistic Neural Network (FWPNN) classification method was developed in [32]. The multiple-criterion feature selection approach was then used to choose 19 highly significant characteristics associated with ALZ disease (AD). The ADNI subset was used for the experimental validation, which was subsequently applied. In terms of three classes of ALZ disease classification, the authors suggested a classification accuracy of around 98.63%. 

Deep convolutional neural networks (CNNs) were used to separate the various ALZ-related diseases. Using transfer learning from previously trained ImageNet (through fine-tuning) and five-fold cross-validation, two CNN architectures, GoogleNet and CaffeNet, were investigated and assessed in numerous classifications and predictions of conversion risk. The CaffeNet attained astounding accuracy levels of 95.42% to 97.01%. With the aim of accurately identifying both AD participants and brain areas associated with AD, we suggested a unique CAD method for MR brain images in [34]. In another study [35], the authors utilized CNN to categorize people based on images of the hippocampal area taken from MRI scans of the brain as having ALZ disease into two classes by using ADNI. From three groups, the prediction model achieved accuracy of 92.3%, 85.6%, and 78.1%.

A new Sobolev-gradient-based optimization with weight values for each decision parameter is presented in [36]. The suggested network model was able to extract necessary characteristics from MRI images and deliver automated diagnosis with 98.06% accuracy, according to experimental findings and quantitative assessments. Classification, extensive network analysis, and support vector machine methods are discussed in [37]. Thus, the author’s approach for identifying Alzheimer’s neurodegenerative disease achieves 98% accuracy. Without any human interaction, the CNNs made their choice in [38]. They ran trials using the ADNI database to confirm the findings of this study, and the results showed that the suggested strategy outperformed previous network models by achieving accuracy levels of 86.60% for the AD classification tasks. The usefulness of texture-based features such as the gray level co-occurrence matrix (GLCM), scale-invariant feature transform, local binary pattern, and gradient histogram is improved in [39] using the bag of visual words technique. By combining clinical characteristics with texture-based features to create a hybrid feature vector, the value of clinical data presented with the imaging data is highlighted. The binary categorization of AD and normal is accurate to 98.4%. An accuracy of 79.8% is attained for the categorization of AD, normal, and mild cognitive impairment (MCI) into several classes. Shi et al. [40] developed a stacked denoising sparse auto-encoder (DSAE) as part of a deep-network-based feature fusion technique to merge cross-sectional and longitudinal data calculated from MR brain images. They assessed the effectiveness of the suggested feature transformation and feature fusion algorithms using the ADNI dataset.

To learn features, they developed a 3D densely connected convolutional network (3D DenseNet) in [41]. Finally, the multi-task CNN and DenseNet models learned features, which are integrated to identify ALZ states, consisting of three classes, using the ADNI dataset. The suggested technique also achieves classification accuracy for AD vs. normal control (NC) patients of 88.9% and an AUC (area under the ROC curve) of 92.5%; for MCI vs. NC subjects, these values were 76.2% and 77.5%, respectively. The framework in [42] linked to a fully convolutional network, which builds high resolution maps of disease probability from local brain structure to a multilayer perceptron to accurately diagnosis ALZ disease. The authors used the ADNI dataset only to train the model. Talo et al. [43] presented a method for automatically classifying normal and pathological brain MRI images. As a DL model, the ResNet34 model, built on a convolutional neural network (CNN), is employed. To train the model, they employed DL approaches such as data augmentation, an optimal learning rate finder, and fine-tuning. On 613 MRI images, the suggested model had a 5-fold classification accuracy of 100%.

A CNN architecture, VGG-16, trained on the ImageNet dataset, is employed in [44] as a feature extractor for the classification job in a mathematical model based on transfer learning. The experiments were conducted with information gathered from the ADNI dataset. For the validation set, the described method’s three-way classification accuracy is 95.73%. Four phases of ALZ disease are classified in the study of [45]. In the first technique, 2D and 3D convolution-based basic CNN architectures are used to process structural brain scans in the ADNI dataset. The authors developed two techniques. The experimental findings demonstrate that the CNN architectures used in the first technique have the following qualities: acceptable basic structures that minimize memory needs, overfitting, and computational complexity; controllable time; and low computational complexity. Additionally, they attain 2D and 3D multi-class AD stage classification accuracies of 93.61% and 95.17%, respectively, which is highly encouraging. The VGG19 pre-trained model has been improved, and it now achieves multi-class AD stage classification accuracy of 97%. Transfer learning was used to train the “Deep Transfer Ensemble (DTE)” in [46] to classify AD. On two separate divisions of the huge dataset for the cognitively normal (NC) vs. AD classification test, DTE achieves accuracy of 99.05% and 85.27%, respectively. DTE succeeds in the job of classifying moderate cognitive impairment (MCI) vs. Alzheimer’s disease (AD) with 98.71% and 83.11% on the two independent splits, respectively.

Lei et al. [47] developed a powerful feature selection model based on group LASSO and correlation and then used a powerful algorithm to optimize it. They tested it using the open ADNI dataset and compared three distinct training approaches. Chen et al. [48] suggested the combination of deep feature extraction and important brain area identification using the iterative sparse and deep learning (ISDL) model to diagnose AD and MCI. According to the findings of their experiments, the ISDL model offers a cutting-edge approach for both AD-CN categorization and MCI-to-AD prediction. Based on structural MRI images that underwent either minimum preprocessing or more thorough preprocessing into modulated gray matter (GM) maps, they employed a traditional support vector machine (SVM) and a deep CNN method in [49]. 

Convolutional neural networks (CNN) [50] and brain MRI scans have been employed in several recent studies, with promising findings for the diagnosis of Alzheimer’s disease. As a result, this study suggests an end-to-end paradigm for classifying AD that is based on CNN. On the Alzheimer’s Disease Neuroimaging Initiative (ADNI) dataset, the suggested framework classified Alzheimer’s disease (AD) cases and cognitively normal cases with accuracy rates of 99.6%, 99.8%, and 97.8%, respectively. The suggested framework obtained 97.5% classification accuracy on the ADNI dataset in multi-classification studies. A unique CNN design that can discriminate between people with normal cognition, MCI, and mild Alzheimer’s disease dementia was described in [51]. The authors used the ADNI dataset to evaluate the CNN architecture. Abuhmed et al. [52] employed a hybrid model architecture to train several machine learning classifiers using deep features. Those features were retrieved from the BiLSTM model. In the research of the Alzheimer’s disease neuroimaging effort, 1371 people participated, and the two architectures were thoroughly assessed using several time series modalities (ADNI). The results of the comprehensive, real-world experiments using ADNI data demonstrate the efficacy and viability of the suggested deep learning models.

According to [53], a deep neural network (DNN) is a machine learning technology with the capacity to take in the most crucial data for effectively categorizing an item. LeNet employs the max pooling layer, like the majority of DNN models, to eliminate the information of least-valued items and reduce dimensionality. Low-intensity pixels in brain pictures may potentially have extremely significant properties. They have added the concatenated layers to LeNet to replace all max pooling layers. The suggested modified LeNet model attained an average performance rate of 96.64%, in contrast to the original LeNet model’s performance rate of 80% for classifying ALZ disease. Hazarika et al. [54] extracted the deep features using an effective transfer learning (TL) architecture to classify AD stages. This research proposed an Alzheimer’s stage identification based on deep features by transferring the initial layers from a pre-trained AlexNet model and extracting the deep features from the convolutional neural network (CNN). For the classification of the recovered deep features, they used the well-known machine learning techniques support vector machine (SVM), k-nearest neighbor (KNN), and random forest (RF). A deep-feature-based model beat both handmade and deep learning methods with 99.21% accuracy, according to the assessment findings of the suggested scheme. Additionally, the suggested model performs better than current cutting-edge approaches.

The CNN-SVM was described in [55] and evaluated using the ADNI dataset. The trials discriminate between people with normal cognition (NC) and those with early-onset or progressing dementia, yielding encouraging findings. Convolutional neural networks and supervised machine learning methods examined achieve accuracy of 92.5% and 75.0% for NC vs. MCI and 93.0% and 90.5% for NC vs. AD, respectively. 

It is suggested to rank the basic classifiers that could stray from conditional independence using deep belief networks [56]. Then, a meta-classifier using a neural network is employed. Under-sampling and threshold-moving are employed at the optimization layer to address the cost-sensitive challenge of recognizing Alzheimer’s disease. Arafa et al. [57] provided the comparative analysis to detect the accuracy of state-of-the-art models for classification of ALZ disease. Compared to other studies, we have used the ConvMixer [58] model to extract deep features and then used a random forest (RF) [59] to classify ALZ disease into four classes. The experiments are performed on the Kaggle-ALZ [60] dataset. In addition, the proposed CAD-ALZ system is compared with [61] and tested on the ADNI [62] dataset.

## 3. Materials and Methods

### 3.1. Data Acquisition

A researcher developed the ALZ dataset [60], which consists of four stages. The collection includes MR images with various resolutions that were gathered from various websites. The hybrid dataset that was used in this investigation was made available online. The collection consists of 5000 JPG-formatted images in total. This dataset was increased by a data augmentation technique which is explained in the coming subsection. In this study, 10-fold cross-validation tests are applied to produce various validation/testing/training ratios. Classes made up of the four phases of ALZ make up the dataset. These stages are mildly (MID), moderately (MOD), normally (NOD), and very mildly (VMD) demented. Furthermore, no comprehensive information was provided regarding the patients from whom the dataset was compiled. Because of this, the homogeneity or heterogeneity of the images in the dataset is unknown. Figure 1a displays examples of MRI images that correspond to the dataset’s classes, and Figure 1b visually displays those sample images of ALZ disease stages.

### 3.2. Preprocessing and Data Augmentation

**Preprocessing:** Images might have noise and distortions in their datasets. Radiography noise is typically brought on by changes in the detector’s sensitivity, decreased contrast due to poor illumination of the object, photographic restrictions, and spontaneous variations in the radiation signal. To enhance the data’s quality or optimize their geometric and intensity patterns, preprocessing is essential. Preprocessing draws attention to the most crucial information required for classification and enables researchers to concentrate on a specific section of the brain. The difference between the greatest and lowest pixel intensities, as defined by Equation (1), is contrast enhancement.

It enhances image quality and raises border contrast in the image, which makes it easier to distinguish between different organs. By extending the range of pixel values, it also enhances the image’s brightness.
(1)$$gray\left(x,y\right)=\frac{I\left(x,y\right)-fmin}{fmax-fmin}\times gray\_levels$$
where *I*(*x*, *y*) represents the value of each pixel in the image, the fmin parameter represents its minimum value, the fmax parameter represents the highest value, and $gray\left(x,y\right)$ represents the enhanced pixel because of applying image contrast. A high-contrast example of an axial MRI brain image is shown in Figure 2. The median filter is a method for reducing noise without introducing edge blur. It is highly suitable for improving the necessary MRI pictures. By comparing each pixel in the image to its surrounding pixels, the median filter can determine which pixels are noise. Each pixel value in the image is passed through the filter (also known as a kernel) and changed to the matching median value. Sorting the values of the pixels in the area yields the median value, which is then used to replace the offending pixel with its matching middle value. Figure 2 depicts the outcome of applying the median filter with a size of (3 × 3) to a sample image.

**Data Imbalance and Augmentation:** Data augmentation is essential for significantly expanding the diversity of data for training neural networks when the starting dataset is small. The overfitting problem is avoided by our model with the use of data augmentation techniques. As a result, this paper used data augmentation for training the proposed classifier. The dataset for this study consisted of 350 patients, which was inadequate to train the deep neural network and enhance performance. Data augmentation can be used after data collection to broaden the variety of images in each class. Various methods exist, including cropping, shifting, shearing, scaling, and zooming. We can expand the dataset’s size using these methods. 

We require enough balanced data to successfully train a DL-based model, as shown in Figure 1a, which shows the bar representation of the class-wise sample distribution of the original dataset. Data balancing is done by purposefully generating the required samples to prevent biased sampling during the DL model’s training. In addition, the imbalanced data may cause the model training to continue favoring classes with many examples. Therefore, to balance our dataset, we used augmentation techniques. The dataset has 5000 total MRI images before data augmentation; after data augmentation, 40,000 MRI images were obtained. As shown in Figure 1, the data is imbalanced. The augmentation strategy and parameters are shown in Table 2. A visual example of the proposed data augmentation approach is shown in Figure 3.

### 3.3. Proposed System Overview

For recognizing multi-class Alzheimer’s (ALZ) disease, this paper has developed the new CAD-ALZ system, which is shown in Figure 4. In the first step, this paper used ConvMixer [57] to extract deep features from the preprocessed dataset and then a random forest (RF) [58] is utilized to classify those features into ALZ stages. The RF classifier is utilized in this paper to categorize AD classes because the RF classifier makes an ideal candidate for handling high-dimensional problems, where the number of features is often redundant. As a result, the RF classifier is utilized to select informative features and to classify the AD disease CAD-ALZ system. The CAD-ALZ model was then tested against the dataset to verify its accuracy. Using the preprocessed dataset, we trained and assessed the CNN deep-learning-based model from scratch. We trained and evaluated the CNN deep-learning-based model from scratch using the preprocessed dataset. In the third experiment, we extracted features from the dataset using ConvMixer, and then we used those features to choose the best classifiers for RF-based deep feature identification. Those steps are explained in the subsequent subsections.

#### 3.3.1. Deep Feature Extraction Using ConvMixer

Compared to ML techniques, the convolutional neural networks (CNNs) require feature extraction and selection steps because prediction relies on those features, which deep CNN’s convolutional layers must learn. Deep CNNs can, nevertheless, self-learn features associated with issues due to defining the size of convolution. The convolutional operations were applied to deep CNNs to automatically extract features. CNN’s initial convolutional layers employ filters to find low-level characteristics like edges, blobs, and colors. The final layers of the networks employ filters to learn higher-level, more complex information. Simple numerical representations of the location of a particular pattern serve as the features that convolutional layers learn.

The wonderful union of CNN and Vision Transformer is the ConvMixer Network [58]. While the ConvMixer separates images into visual embeddings, CNNs use pixel arrays. After dividing an image into fixed-size tokens, the visual transformer sends positional embedding as an input to the transformer encoder. Basically, it portrays an image as a collection of words or word embeddings (patches). Instead of using pixels like a CNN, it separates the image into tokens or patches. This raises the intriguing question of whether its improved performance was a result of the transformer architecture or the use of patches rather than pixel arrays. The ConvMixer illustrates the latter argument and, because of this, invents a straightforward but incredibly powerful network.

The pre-convolutional block, convolution-mixer block, and post-convolutional block are the three key components of the study’s ConvMixer networks. To extract deep characteristic features, these blocks are employed. This paper used a model encoder on depthwise separable convolution since it offers the best computing performance with a minimal amount of model parameters. With the same neural layers of a two-dimensional ConvMixer, batch normalization, and the Gaussian error linear unit (GELU) activation function, this study created pre- and post-convolutional blocks.

ConvMixer receives the patches and separates the spatial and channel mixing while ensuring that every patch has the same size and resolution throughout the network. Convolutions are used, though, to accomplish the mixing processes. Additionally, batch normalizations are used in place of layer normalizations. It performs significantly better than both its rivals and the most fundamental CNN models, despite using ordinary convolutions. The ConvMixer model, which is nothing more than a convolution layer with c input channels and h output channels, kernel size, and stride equal to the patch size, is fundamentally composed of a patch-embedding layer. An activation function and batch normalization are then used following the activations. This is the initial element of the model.
(2)$$Zo=BN\left(\sigma \left\{{Conv}_{cin}\to h(x,stride=p,ksize=p)\right\}\right)$$

In Equation (2), the *BN* is a branch normalization, the *ksize* parameter is the kernel size of patch (*p*), stride *p*, σ is the GELU activation function, Zo is the parameter outputted from the ConvMixer module, and *h* is the patch-embedding dimension. The second component of the model is the primary ConvMixer layer, which is composed of depth times that are repeated. A depthwise convolution has been applied to the residual blocks that make up this layer. Simply explained, a residual block is one that multiplies the result of one layer by the result of another. The depthwise convolution (${Conv}_{Depth}$) output layer in this instance is created by concatenating the inputs. An activation block, a pointwise convolution (${Conv}_{point}$), and another activation block are placed after this output.
(3)$$Z_{l}^{1}=BN\left(\sigma \left\{{Conv}_{Depth}(Z_{l-1}^{1})\right\}\right)+Z_{l-1}^{1}$$
(4)$$Z_{l+1}^{1}=BN\left(\sigma \left\{{Conv}_{point}(Z_{l}^{1})\right\}\right)$$
where the parameters ($Z_{l}^{1}$, $Z_{l+1}^{1}$) are outputted from the ConvMixer module. A global pooling layer, which makes up the third part of the model, creates a feature vector of size *h* that can be given to a GELU (σ) or any other head, depending on the job. Throughout the entire model, the activation function known as GELU, or Gaussian error linear unit, is employed. The GELU activation ($GELU\left(x\right)$) function, in contrast to RELU, weights inputs according to their magnitude rather than gating them according to their sign.
(5)$$GELU\left(x\right)=x.\varphi (x)$$

As shown in Figure 5, the subsequent blocks are all convolved with various kernel sizes and padded to maintain the dimension from the prior time interval. The property of translation equivariance for the convolutional operation in 1D is not preserved in the frequency domain, according to [16], which was discussed. The learning of some geographical information as well as the frequency channel would be compromised by this. As a result, we are thinking of implementing 2-dimensional depthwise separability, particularly in our ConvMixer block. The frequency domain extraction process in the ConvMixer block uses the time feature from the previous channel as input. Through this, the rich information from the frequency domain is expressed in a third dimension. We used a pointwise convolution to compress the new input back into the old input’s shape to preserve the shape of the previous input. Then, using a 1-dimensional depthwise separable block, we put the temporal domain feature extraction into practice. Frequency- and temporal-rich embeddings will be the end outcomes of these two processes. Then, to facilitate the transmission of data via the global feature channel, we constructed a mixer layer. The 2D features of connecting to the block output and skipping connections from the preceding output were implemented last.

The strength of the attention layer, which enables networks to concentrate on relevant spatial information, has made it popular. However, this necessitates extensive linear computation. The authors in [17,18] proposed mixing the tokens channel-wise as an alternate method of feature transmission to weighing the significance of each element in relation to each other token. To create the interaction between the feature space and the environment, we suggested using two different forms of multi-layer perception (MLP), namely temporal channel mixing and frequency channel mixing. Two linear layers and a GELU activation unit, which are independent of each temporal and frequency channel, are used in each MLP mixing. This is defined as:(6)$$u_{n,i}=x_{n,i}+w_{2}\times \delta (w_{1}\times L\_Norm(x{)}_{n,i})$$
(7)$$y_{n,n}=u_{j,n}+w_{4}\times \delta (w_{3}\times L\_Norm(x{)}_{j,n})$$
where the *δ* function represents the GELU unit and the $L\_Norm\left(.\right)$ function shows the linear layers. The *W1*, *W2*, *W3*, and *W4* are the learnable weights of the linear layers for frequency channel shared across all *j*, for *j* ∈ {1, *J*}. To get more informative features, the process for the multi-channel ConvMixer repeats for a total of N = 8 times. The parameter N = 8 is fixed, and it is determined after doing experiments. The Algorithm 1 displays all summarized steps to develop the ConvMixer model for the extraction of effective features from MRI images.

**Algorithm 1:** Proposed ConvMixer model with fine-tuning for robust feature extraction
1.**Input:** Input Training Data, Tensor (X), Image patches.2.**Output:** Extracted feature map x=(x1,x2,.....,xn)3.**Main Process:**Function Depthwise_CNN and Pointwise_CNN for depthwise separable convolution.4.block (b1) = Depthwise_CNN and Pointwise_CNN function block takes as inputs tensor (X).5.- For each ConvMixer block do6.- Depthwise_CNN function includes (3 × 3) size is applied to X inside Depthwise_CNN function. The BN function is performed. ReLU() activation function is applied and Pointwise_CNN() function including (1 × 1) size is applied to X inside Pointwise_CNN function.BN() function is performed. ReLU() activation function is applied.**[Blockwise fine-tune strategy for pre-trained model]**7.- (a) Remove last FC and Softmax layer from Inception v3 and Add Dense layer and dropout layer8.- (b) Remove last FC and Softmax layer from Inception v3 and Add dense layer to network- [End for loop]9.Structure of Model
(a)Start with (3 × 3) size of two Conv layers, each with 32 and 64 filters, followed by ReLU function.(b)Next, residual network based on three skip connection is added to the network.(c)Each skip connection includes two separable Conv layers followed by max pool.(d)The skip connection has Conv of 1×1 with strides of 2.
10.Afterward, the feature map F $=\left\{f1,f2,\ldots ,fn\right\}$ is generated by using average pool layer.

#### 3.3.2. Proposed ConvMixer Model Architecture

Figure 5 depicts the conceptual framework for the suggested CNN-based ConvMixer architecture. The proposed ConvMixer recognizes four class labels from a (256 × 256) input MRI image. One GELU activation function with a branch normalization layer, eight ConvMixer blocks to extract deep features, and finally a global average pooling layer make up the proposed architecture. To extract spatial connection between feature maps and ultimately transmit it to the random forest (RF) classifier for classification, the convolutional blocks include eight designated blocks. These blocks consist of:

(1) Convolution Layer: To limit the number of parameters, we employ small convolution kernels (e.g., 3 × 3) in the proposed CNN rather than big convolution kernels (e.g., 5 × 5). The tiny convolution kernels can efficiently extract local characteristics while reducing the number of parameters.

(2) Batch Normalization Layer (BN): During the training, the distribution of each mini-batch is typically normalized using batch normalization to a zero mean and a unit variance. Utilizing a BN layer has the benefit of successfully preventing gradient vanishing or explosion and overfitting in ConvMixer, as well as enabling a reasonably high learning rate to hasten convergence. We discovered via the tests that networks without BN, like Ye-Net, are particularly sensitive to parameter initialization and may not converge with insufficient initializations. As a result, in the suggested strategy, we employ BN.

(3) Non-Linear Activation Function: To avoid gradient vanishing or exploding, creating sparse features, hastening network convergence, or achieving other objectives, we employ the advanced Gaussian error linear unit (GELU) compared to the traditional rectifying linear unit (ReLU) as the activation function. GELU can be used on neurons to train them to react only to inputs that contain relevant signals, leading to the development of more effective features.

(4) Depthwise and Pointwise Convolution: A separable convolution in depth divides the operation into two parts, such as a depthwise and a pointwise convolution. In the first section, known as depthwise convolution, we apply a convolution to the input image of size (256 × 256) without altering its depth. Because it utilizes a 1 × 1 kernel as a kernel that iterates across each point, this step is known as pointwise convolution. This kernel has a depth equal to the number of channels present in the input image.

#### 3.3.3. Blockwise Fine-Tuning Strategy

First, each of the pre-trained models is fine-tuned using the blockwise method so that the CNN model can manage the heterogeneity in the colon histopathology pictures. Additional abstract features are extracted from the image to improve intra-class discrimination. Second, the performance of the four adaptable pre-trained models is enhanced through ensemble learning. The final judgment on the test images will be more accurate as a result.


In CNN, each layer is regarded as an independent block, as seen in Figure 5. From block 1 (B1) through block 8, there are eight blocks (B8). These blocks are from fully connected layers to the image input layer. This model is tweaked block by block, starting with B1 and progressing through B1, B2, B3, B4, B5, B6, B7, and B8. Based on determining the learning rate for the relevant block layers, block-by-block fine tuning is carried out. As an illustration, the learning rate is initially set to a fixed value for block Bx, and it is zero for blocks Bx-1 through Bx-n, where n = x − 1 and x is the number of blocks. The learning rate for blocks Bx and Bx-1 is set to a fixed value in the following phase, while it is zero for blocks Bx-2 through Bx-remaining n’s blocks. This procedure will continue until all blocks are properly calibrated. Algorithm 1 shows the blockwise fine-tuning CNN model.

To categorize the images of two histopathology datasets using three learning rates, we compared the blockwise effect of fine-tuning three cutting-edge CNNs. According to the study’s findings, fine-tuning only the top blocks of the network rather than the entire network produced the highest performance for Inception v3 architectures. The performance of the Inception V3 architecture was improved by tweaking the entire network. The bottom layers’ generalizability, or the ability to utilize them with any type of dataset and have their weights fixed, is the key point of contention.

### 3.4. Feature Classification by Random Forest (RF) Classifier

ConvMixer-based deep features are extracted from Section 3.3.1 and trained by a random forest (RF) [59] classifier to predict classes of Alzheimer’s disease. From each image, this study extracted 4096 features. These features are fed into an RF classifier for training. In practice, using RF for ALZ disease classification reinforces the reliability of the prediction since an ensemble of decision trees is employed. The RF classifier is a type of ensemble-based learning technique. In this study, the RF is selected because it overcomes the problem of overfitting by averaging the results of different decision trees. Furthermore, it is very flexible and possess very high accuracy. There are many standard machine learning classifiers available, such as neural networks (NN), support vector machine (SVM,) adaptive boosting (AdaBoost), Nave Bayes, and RF. However, this paper applied the RF classifier because this classifier used a combine approach by using bagging and random feature subset sampling to construct the classification trees. Therefore, the RF classifier produces multiple models to make accurate predictions compared to a single model alone. To construct a subset of training data, the RF model uses a random subset of features to learn each tree in a forest. About one-third of the training data for each tree is left as “out-of-the-bag” data after a subset of the training data is randomly sampled with a replacement strategy. As new trees are added to the forest, the out-of-the-bag data are utilized to calculate the classification error rate and can be used to gauge the significance of a feature. A test sample is classified using most of the votes from all of the outputs from all of the models. The number of trees and maximum characteristics for each split are two hyperparameters for building an RF classifier.

Using weak decision trees in parallel, this paper uses the majority vote to integrate the trees into a single, powerful learner. The RFs are frequently discovered to be the most precise learning algorithms in use today. Algorithm 2 provides an illustration of the pseudocode. The procedure is as follows: we choose a bootstrap sample from t, where t(i) stands for the ith bootstrap, for each tree in the forest. Afterwards, a modified decision tree learning algorithm is used to learn a decision tree. The procedure is changed in the following way: at each node of the tree, we randomly select some subset of the features f ⊆ F, where F is the collection of features, rather than looking at all potential feature splits. In practice, F is much smaller than f. Sometimes the most computationally expensive part of decision tree learning is choosing which feature to split. We significantly accelerate the learning of the tree by restricting the number of features.

An RF classifier is used to train and categorize the features that were retrieved using the techniques described in Section 3.4. This work utilizes the Scikit-Learn library package to create an RF classifier. Except for the classifier’s number of trees, all the RF’s parameters are set to their default values. The first test on the normal versus Alzheimer’s disease problem was undertaken using relatively few features and many features to test the greatest number of trees possible. Before the RF classifier, no feature selection or feature reduction is carried out.
**Algorithm 2:** Random Forest (RF) classifier and final decision with majority voting scheme
1.**Input:** Input A training set t=((x1,y1),…,(xn,yn)), features F=f1,f2,…,fn, number of trees in forest (m).2.**Output:** Predict class for Alzheimer’s disease.**Main Process:**3.function RandomForest (tn, Fn)4.Initialize H=∅ and For i = 1 to m do5.ti=A bootstrap sample from t training set6.hi=RandomizeTreeLearn(ti,Fn) and H=H∪{hi}[end for loop][Function end]Function RandomizeTreeLearn (tn,Fn)7.If node contains only one class, then return else8.At each node of tree: f=very small subset of F and split on best information gain features in f.9.Return RandomizeTreeLearn (tn,fn)10.[End function]

## 4. Experimental Results

### 4.1. Environment Setup

The suggested deep-learning-based technique is evaluated in this study using a variety of training and validation datasets. Datasets are divided into a training set and a testing set in every experimental session using a 3:1 split, which designates that three-quarters of the data are utilized for training, and the remaining one-quarter is used for testing. All photos were scaled to (256 × 256) pixels to carry out feature extraction and classification activities. Our suggested CAD-ALZ system is built by combining the ConvMixer model to extract features with a random forest (RF) classifier to categorize those features. To construct and program the CAD-ALZ system, a computer with a core i7 CPU, 16 GB of RAM, and a 4 GB Gigabyte NVIDIA GPU was used. On this computer, TensorFlow (version 2.7) and Keras deep learning (DL) libraries are set up using Windows 10 Professional 64-bit edition.

The CNN architecture is built and trained using a variety of kernel dimensions in an organized manner to produce feature maps from the previous step. The convolutional layer’s weight values are altered since the project typically uses kernel dimensions of either (3 × 3) or (5 × 5). Various window widths and values obtained from the excitation objective function of each feature map are used to convolute the convolutional layers. The pooling layer was made using a similar procedure to the convolutional layer. There is only one difference: a window size of (2 × 2) and sliding increments of 2 are utilized to maximize the features gathered from the previous layer. This stage reduces the convolutional weights while increasing the network’s overall speed. This average pooling’s output is passed into a fully linked RF classifier. The four phases of ALZ disease are distinguished using this FC stage. The number of parameters in the proposed model’s convolution layer is shown in Table 3.

### 4.2. Statistical Performance Metrics

The performance of the suggested CAD-ALZ was thoroughly assessed using the Kappa score (Kappa), sensitivity (SE), specificity (SP), precision (PR), accuracy (Acc), and F1-score metrics. The six metrics’ formulae are as follows:$$Ke=\frac{\left(TN+FN)\times (TN+FP)+(TP+FP)\times (TP+FN\right)}{\left(N\times N\right)}$$
$$Ko=\frac{\left(TP+TN\right)}{N}$$
where the Ke and *Ko* parameters are defined by using the data that have already been collected to figure out how likely it is that each observer is randomly perceived in each category. Therefore, the *Ko* parameter is the proportion of raters who agree, and the Ko parameter is the chance that they will agree. For Ke and Ko, the *Kappa* score is defined as follows:(8)$$Kappa=\frac{\left(Ko-Ke\right)}{\left(1-Ke\right)}$$

The sensitivity is defined as:
(9)SE=TP/(TP+FN)

The specificity is defined as: (10)SP=TN/(TN+FP)

Precision is defined as: (11)PR=TP/(TP+FP)

Accuracy is defined as:(12)ACC=(TP+TN)/(TP+TN+FP+FN)

F1-score is defined as:(13)F1−score=2×PR×SE/(PR+SE)

The terms true positive, true negative, false positive, and false negative, respectively, are represented by the letters TP, TN, FP, and FN. Kappa, SE, SP, PR, ACC, and F1-score were only a few of the six assessment measures that were used to gauge how well the proposed CAD-ALZ system and other cutting-edge methods performed. The Kappa is a statistical indicator of the model’s predictability. The SE is a crucial diagnostic statistic that is linked to the success rate of the prediction. The SP is essential for determining the correctness of the model and for identifying medical issues. The PR describes the model’s capacity to make a successful prediction. The accuracy of the model’s prediction is indicated by the Acc. The harmonic is the F1-score of the PR and SE.

### 4.3. Results Analysis

Using 2D MRI dataset images, the performance of the suggested CAD-ALZ was examined to evaluate the model’s generalizability and transferability using six metrics (Kappa, SE, SP, PR, Acc, and F1-score). To assess the effectiveness of the suggested CAD-ALZ system, two sets of experiments were carried out. The empirical distributions of the boxplots were computed using the bootstrapping technique. The same dataset was used in all trials. Ablation research was also carried out to show more clearly how effective the suggested framework is. The proposal refers to the proposed framework as illustrated in the figure and shown in Table 4 and Table 5.

Using the PyTorch framework, the training and inference procedures were carried out. The dataset is divided into various testing, validation, and training ratios by using k-fold cross-validation. The k = 10 parameter is used to assess how well the trained model performed. The training was repeated ten times. This paper divides the data into 10 pieces, each 10% of the full dataset. In each iteration k, the training set consisted of data from nine out of the ten folds. As a result, we used the first fold in the first experiment as a validation set and used everything else as training data. In the second experiment, we keep data from the second fold (and use everything except the second fold for training the model). We repeat this process, using every fold once as the validation set. Combining all this together, we obtained 100% of the data used as the validation test.

The performance of the suggested CAD-ALZ system was examined using samples from the training dataset. Figure 6 is used to display the training and validation accuracy versus loss of the proposed CAD-ALZ system with preprocessing and data augmentation steps, whereas in Figure 7 is shown an example of the confusion matrix of the proposed CAD-ALZ system to recognize all stages of AD (a) without a preprocessing step and (b) with a preprocessing technique.

Results of the dataset are measured based on 2D MRI scans. These images were utilized in this experiment to assess how well the models performed. Table 4 lists the overall outcomes of the six metrics for the proposed CAD-ALZ system compared to other pre-trained models such as VGG-16, VGG-19, and AlexNet architectures. With an F1-score of 99.61%, CAD-ALZ has the best performance, followed by AlexNet (98.67%). The Acc for CAD-ALZ was greatest (99.61%), followed by VGG-19 (98.36%) and AlexNet (98.60%). The PR, SP, SE, and Kappa of the CAD-ALZ were, respectively, 99.53%, 99.53%, 99.69%, and 99.22%. The index values exceeded those of CAD-ALZ (98.60%, 98.59%, 98.75%, and 97.34%) compared to others.

Using data augmentation techniques, the initial dataset was reproduced in the experiment’s second phase. The Vit, DSC, and VGG-16 models were each trained on three datasets. The NN and SVM techniques were used to categorize the deep characteristics that the VGG16 model gave. The dataset produced using NN had an overall accuracy success rate of 82.0%. The CAD-ALZ with RF classifier technique’s overall accuracy success rate was 98.94%. Figure 6 displays training accuracy graphs for data enhancement, and Figure 7 displays confusion matrices for the proposed CAD-ALZ system. Datasets were trained with the VGG-16 model in the experiment’s third step, just as they were in the second. The FC-7 layer of the VGG-16, which has 4096 deep features, was taken into consideration. A new feature set with 12,288 features was produced by combining three feature sets generated using the VGG19, vision transformers (Vit), and depthwise separable CNN methods. The NN and SVM were then used to classify the merged feature collection. The test data were set at 25% in this case, just like in the preceding steps. The overall accuracy rate discovered through the classification was 99.6%. The third step’s achievement produced a better outcome than the first two steps did. However, because the dataset’s feature count (12,288) contains too many features, a reclassification using effective features was required.

### 4.4. Computational Cost

It took 25 s to complete the suggested picture preparation step, which altered the input image’s brightness and eliminated noise. Similar to this, the suggested CAD-ALZ system’s feature learning and extraction stage took an average of 16.5 s, but the random forest (RF) classifier’s creation time was just 3 s. Thus, on a fixed number of iterations, the RF training for the binary classification of Alzheimer’s disease into normal and dementia classes took 5.20 s. However, categorizing the image only needs an average of 8.12 s once the training is finished and the test is done. The convolutional neural network (CNN) model was used to calculate the suggested CAD-HR, which took 2.1 s longer to compute than [7,33]. This is due to the fact that we used DSC, three residual blocks, and RF to execute a revolutionary approach.

In order to select the model that is most appropriate for your use case, you must have a basic understanding of how to gauge a model’s speed. In this study, we compute (1) FLOPS, (2) number of parameters, (3) frame rate, and (4) delay using two straightforward scripts. You may assess the model’s speed using these four numbers. FLOPS stands for floating-point operations per second, while fps stands for frames per second. In terms of comparison, (1) FLOPS, (2) number of parameters, (3) fps, and (4) latency, the lower the better. The lower all of these, the better. We use the input option found in each model’s training configuration. For instance, transfer learning models are required for stride 2 in crop size 112 compared to crop size 224. It is ensured that the speed performance here and the stated accuracy number are closely correlated. By the proposed model, the FLOPS of 198.874, the number of parameters of 71.800, the fps of 716.89, and the latency of 0.0893 are observed.

Comparatively, the ConvMixer-V8 model is an efficient tool for categorizing ALZ disease with a better computational cost. The original ConvMixer network contains more parameters and FLOPS (given in Table 6) compared to the suggested ConvMixer-V8 model than other CNN- and TL-based designs. As a result, the upgraded architecture converges more quickly than its baselines and has fewer parameters. Table 7 displays the FLOPS (67.3 M), parameter (1.9 M), model size (9.3 MB), and GPU speed (0.6 MS) values. The ConvMixer-V8 model, as a result, produced an updated and redesigned architecture, which is explained in Section 3.

### 4.5. State-of-the-Art Comparisons

The fourth step’s accomplishment (state-of-the-art comparison) was more impressive than the other three steps, such as preprocessing, feature extraction using ConvMixer with a fine-tuning strategy, and classifying those features using random forest (RF). Additionally, the fourth step’s selection of effective qualities helped to support the suggested method. The proposed technique produced classification results for the mildly demented class that were 100% successful, 99.94% successful for the moderately demented class, 100% accurate for the non-demented class, and 99.94% accurate for the very mildly demented class. Those results are already mentioned in Table 4 and Table 5.

To perform comparisons with state-of-the-art methods, this study used Murugan-DEMNET-CNN [15]. For the DEMNET-CNN [15] technique, we utilized the CNN architecture to detect two classes versus four classes in ALD. A Siamese-CNN neural network is developed in this paper [16]. The Islam-Deep-CNN [17] used a CNN architecture based on convolution, batch normalization, rectified linear unit, and pooling. The Janghel-VGG16-SVM [19] was also used to perform comparison based on a pre-trained VGG-16 architecture for feature extraction and SVM to classify AD.

Whereas the Talo-ResNet-50 [21] used a transfer-learning-based architecture such as ResNet-50 to classify AD disease, the Gunawardena-CNN-SVM [23] used CNN and SVM classifiers to detect AD disease. Wang-CNN-Maxpool [26] used classification based on an eight-layer CNN model with leaky rectified linear units and maximum pooling. These studies were selected because they are easily implemented and can be compared with the proposed deep-learning architecture based on ConvMixer and the RF classifier. The corresponding results in terms of confusion matrices are shown in Figure 8. Compared to other approaches, the Talo-ResNet-50 [21] shows a higher classification result for AD in two stages (demented and non-demented) and four stages (very mild, moderate, mild, and non-demented). In addition, Figure 8f shows another study that outperformed the first, which was based on the Gunawardena-CNN-SVM [23] system to detect AD disease in terms of two stages and four stages. Accordingly, those results demonstrate that the proposed technique produced better classification results than state-of-the-art approaches.

Table 5 provides the metrics for the confusion matrix’s third and fourth phases. To avoid issues like over- or underfitting, in the last stage of this work, the suggested strategy was validated using the cross-validation method. The cross-validation coefficient’s value (k = 10) was set to 10. In the last step, cross-validation analysis and the RF approach were used to assess and categorize the combined feature set (12,288 features). The total accuracy success percentage of the categorizing procedure was 99.66%. The classification procedure took 2292 s to complete. Then, the cross-validation approach was used to process and classify the 1000-feature set that was chosen using the deep feature selection method by using a classifier.

Overall accuracy success was achieved in the classification process, with 99.69% accuracy. There were 39 s that passed during the categorization procedure. Figure 7 displays the fifth step’s confusion matrices. Furthermore, Figure 8 provides the analysis findings from the cross-validation procedure. In this confusion matrix, the accuracy success demonstrated the effectiveness of the suggested strategy. The cross-validation procedure has proven the proposed approach’s stability. In Figure 8, a comparison between the proposed ALZ-CAD framework and other relevant cutting-edge studies is made. There is little doubt that the ALZ-CAD framework performs better than most of the related research. Designing a broad framework that makes use of the ConvMixer layer and a blockwise fine-tuning strategy to extract deep features is one of the key goals of the recommended methodology. The recommended framework’s insensitivity to datasets and their outliers, as explained in Section 3.4, is one of its key advantages. 

### 4.6. Analyze Generalizability of CAD-ALZ System

In another experiment, we compared the CAD-ALZ system with [61] by using another dataset in ADNI [62] to show the generalizability of the proposed technique. The “A3C-TL-GTO” framework was developed in [61] for classifying ALZ into four stages, and the experiments were performed on the Kaggle-ALZ [60] and ADNI [62] datasets. The suggested approach reduces the bias, variability of the preprocessing steps, and optimization hyperparameters for the classifier model and dataset. The experimental findings show that the “A3C-TL-GTO” framework achieved 96.25% accuracy for the ADNI dataset and 96.65% accuracy for the ALZ dataset based on MobileNet and Xception classifiers. In fact, the authors optimized hyperparameters of TL architectures by using the Gorilla Troops Optimizer (GTO). In general, the authors used the pre-trained TL model and metaheuristic optimizers to fine-tune the hyperparameters. From ADNI, the data are partitioned into three classes, AD (Alzheimer’s), NC (Normal Cohort), and MCI (Mild Cognitive Impairment), and organically counted, 17,976, 138,105, and 70,076. After data augmentation, they are balanced classes. The data augmentation step is applied as mentioned in Section 3.

On average, the proposed CAD-ALZ system as shown in Figure 9a outperformed (SE of 93%, SP of 96%, 94%, ACC of 0.95, PR of 0.98, Kappa of 0.99, F1-score 99%) on five classes of Kaggle-ALZ compared to A3C-TL-GTO-Xception (SE of 87%, SP of 88%, ACC of 89%, PR of 0.88, Kappa of 0.87, F1-score of 88%) and A3C-TL-GTO-MobileNet (SE of 87%, SP of 88%,ACC of 89%, PR of 0.88, Kappa of 0.87, F1-score of 88%). In the case of Figure 9b, the ADNI dataset is used to classify ALZ into three classes. On average, the proposed CAD-ALZ system outperformed (SE of 93%, SP of 96%, 94%, ACC of 0.95, PR of 0.98, Kappa of 0.99, F1-score 99%) on three classes of ADNI compared to A3C-TL-GTO-Xception (SE of 88%, SP of 89%, ACC of 90%, PR of 0.89, Kappa of 0.88, F1-score of 89%) and A3C-TL-GTO-MobileNet (SE of 89%, SP of 90%, ACC of 90%, PR of 0.89, Kappa of 0.89, F1-score of 90%). Those statistical results indicate that the proposed CAD-ALZ system is effective and provides a generalized solution for the classification of four stages of ALZ disease. A visual example of Figure 10 shows the result produced by our proposed CAD-ALZ system.

## 5. Discussion

A trained CNN model was used as an input to DSC architectures to build CAD-ALZ which classifies the ALZ illness using three successive residual blocks and an RF classifier. By creating a multilayered hierarchical structure, the learning process for specialized features was completed without the need for complicated feature selection and image processing techniques. The learning algorithms are used to immediately learn this multi-layer architecture from the input image, negating the requirement for human interaction. DSC and residual blocks are added to the Xception model to produce more generalizable features for CAD-ALZ architecture development. The depthwise convolutional layer obtains localized and learned features based on a scratch-based training method. Convolutional, pooling, and fully connected layers make up many of the layers in the CNN model for learning deep features. These layers must first be trained and proven to be successful at obtaining meaningful information before being used to build the model. These characteristics are not the best for using MRI scans to identify Alzheimer’s disease stages. In contrast to feature-based categorization techniques that needed human involvement, deep residual linkages were incorporated, thereby providing highly specialized characteristics. 

An independent feature-learning technique enabled the success in diagnosing ALZ illness. The handcrafted-based categorization systems for diagnosing ALZ disease, however, need computationally costly algorithms for the preprocessing and localization of ALZ-related data. Several categorization schemes for ALZ have been developed in the past, as discussed in Section 3. Instead of using conventional machine learning methods, these systems employ deep learning approaches. There were certain significant challenges when ALZ automated systems were constructed using traditional methods. The first issue arises from the requirement to utilize complex pre- or post-image processing methods to identify and extract pertinent information from MRI images to establish precise parameters. As a result, it is difficult for automated systems to recognize the symptoms of these disorders. The authors trained the network with manually created characteristics and evaluated the effectiveness of both conventional and state-of-the-art deep learning models in accordance with the literature. Consequently, to find the best characteristics, an automated method is required. Deep-learning models produce better results than the traditional method. Other models, however, used trained models created entirely from scratch to automatically learn features, and they all applied the same weighting methodology at each level. It could thus be challenging for layers to communicate precise decision-making weights to deeper network levels.

We created a CAD-ALZ approach for the detection and classification of MD in 2D MRI multimodal brain imaging data utilizing ConvMixer DL algorithms. Based on six assessment variables, the suggested CAD-ALZ performed better than prior state-of-the-art approaches and showed good performance. A backbone and task-subnet modular structure make up the proposed CAD-ALZ. To maintain the functionalities while slowing framework deterioration, we added a residual block to the backbone network architecture. To ensure that the backbone network has good discriminating ability at a low computational cost, we implemented a blockwise fine-tuning technique.

Due to its excellent performance and reliable stability, the CAD-ALZ has a promising future as an adjunctive tool to assist in the diagnosis of MD. Furthermore, it is challenging to locate sufficient and high-quality annotations since real-world medical diagnosis is far more complicated than in experimental settings. Therefore, the creation of our suggested ConvMixer-based DL algorithms is essential for identifying ailments like MD disease. However, when the data distribution is notably imbalanced by category, the suggested CAD-ALZ may run into issues. It is critical to develop effective generation frameworks that can produce a wealth of valuable data for ConvMixer-based compensation solutions.

Over time, MD symptoms get worse, but the disease does not advance in the same way. The average lifespan of a person with mental decline is four to eight years [26]. This study used brain MR imaging of AD to assess the various phases of dementia in Alzheimer’s disease patients. For experts, identifying an Alzheimer’s patient’s phase could be challenging. Here, we propose combining the deep learning model with methods for image processing to enhance expert judgment and simplify this challenging procedure. Our goal was to improve the study’s Inception v3 deep learning model’s performance. Other convolutional models might be used in this case instead of the Inception v3 model. The Inception v3 model has a 96.31% success rate in its initial dataset. In the datasets we improved using the CAD approach, we raised this success to 98.94%. Additionally, it has been demonstrated that the technique helps increase the proposed approach’s prediction success. In this case, we saw that the CAD-ALZ approach fell short of our expectations. Nevertheless, we continued to incorporate it into the experiment’s procedures.

In the last stage of the trial, we considered the possibilities of providing effective features. Therefore, the strategies made a big difference in the experiment. Furthermore, the features produced from the initial image data were absent from the combined feature sets. The objective in this case was to effectively categorize the MD phases using the acquired deep features. The experimental analysis of this study showed that using data-boosting techniques and methods, MR images produced successful results in the detection of MD stages. MD causes near-forgetfulness in some people and dementia in others because of the loss of cognitive functions in the brain. The prevalence of MD is constantly increasing. The stages of MD were identified using the suggested methodology and the information gathered from MD patients. Using this method, MRI images are enhanced, which is thought of as preprocessing. To develop this CAD-ALZ system, it was observed that the preprocessing step was also useful.

Understanding whether dataset boosting affects model performance was one of the main goals. The findings supported the validity of this subject. Additionally, choosing more effective features by merging feature sets is a key goal of this study. Overall accuracy once this process was completed was 99.94%. The proposed method, which was based on the data-enhanced MR images in this investigation, showed promise in identifying the stages of AD. However, the lack of statistical data regarding the patient information of the images in the dataset casts doubt on the precision of the strategy we advise. The researcher who provided the data did not make any comments regarding this circumstance.

Figure 9 shows the visual example of detected ALZ disease with different classes on the selected dataset. As a result, we paused and considered the possibility that the same Alzheimer’s patient may have provided many images. This circumstance may have led to overfitting learning, which can boost the suggested approach’s training and testing success. Even though we minimized overfitting learning by using the cross-validation approach at the end of the experiment, this situation does not entirely remove the dataset’s hesitations. Additionally, the dataset’s images are of poor quality, which could hinder the effectiveness of the suggested method. By utilizing attention modules (regional emphasis) on images, we intend to enhance the performance of the suggested approach on various datasets in the upcoming study. In the future, the CAD-ALZ method for identifying Alzheimer’s disease can be improved by providing a larger collection of retinograph pictures that have been obtained from diverse sources. It might be possible to integrate hand-crafted features to increase the model’s classification accuracy rather than just using deep features. 

### 5.1. Limitations of CAD-ALZ System

Several MRI studies classifying Alzheimer’s disease (AD) were developed in the past by using machine learning and deep learning. However, there are several advantages of the proposed CAD-ALZ system in terms of performance and results, but there are also a few limitations of the proposed system, such as:Lack of changes in a patient’s state of health between a pair of trips. Out of the total MRI images gathered during patient visits, there are only few transitions. Instead of generalizing the crucial differences between the various phases of ALZ disease, it is simple for the model to overfit and memorize the status of a patient at each visit.The time complexity of the CAD-ALZ model can be calculated further on the cloud platform.Our goal is to make models that can accurately classify brain MRIs based on their true stage in the progression of Alzheimer’s disease. These models should be able to do this regardless of the number of visits, the lack of patient transitions, or small differences in scan quality. We hope to keep these difficulties in mind when designing future experiments.We can test our proposed transfer learning model to other datasets, which would make it easier to increase these models’ capacity to generalize across datasets.The preprocessing step is applied to increase the contrast and reduce the noise. We should test different methodologies compared to what has been used in this paper.

### 5.2. Future Directions of CAD-ALZ System

To our knowledge, none of the prior research has made use of every MRI contained in the Kaggle-ALZ and ADNI datasets, and they have not provided a convincing justification for the classification of four stages of ALZ disease. To address the problem, we will conduct our experiments using all accessible data, and we will publish the participants utilized in the training and test split of all our studies for repeatability.

The ubiquitous accessibility of GPUs has been one of the major factors in the rapid emergence of deep learning. GPUs have parallel computing engines that outperform CPUs in terms of execution thread power. Deep learning is known to run 10–30 times faster on GPUs than on CPUs. The abundance of open-source software tools is another factor contributing to the adoption of deep learning techniques. The most popular deep learning libraries are Caffe, PyTorch, TensorFlow, and Theano. An overview of the hardware and software will be needed for DL-based architecture for brain MRI categorization of ALZ disease.

## 6. Conclusions

Alzheimer’s disease detection (AD) is a challenging problem that must be solved to increase accuracy. In this paper, we developed a new CAD-ALZ method for classification of the four stages of AD based on a ConvMixer blockwise fine-tuning approach and random forest (RF) technique to address the problems of the limited dataset, high computational complexity, and unavailability of lightweight and efficient feature descriptor. Utilizing MRI images, this CAD-ALZ model automatically extracts useful characteristics. The proposed network’s input feature map is then classified into four phases of AD illness using the RF classifier. The Kaggle-ALZ database and the ADNI dataset are used to perform a thorough evaluation of the proposed system. The experimental findings demonstrate that our suggested approach outperforms the current deep learning model in terms of performance enhancement and computing cost, both of which are crucial for the implementation of real-time ALZ illness diagnosis. In this paper, we improved and modified these models so they would function properly for our issue. Modern techniques are utilized as feature extraction and classification. For the deep feature, we obtained the best accuracy of 99.21%. Results suggested that research using the ConvMixer model performed better than that using other approaches. These networks’ results represent the pre-training of these models using a large dataset, which is how they were able to attain the highest levels of accuracy. Results from our suggested models were fairly encouraging, since the proposed CAD-ALZ system has obtained high accuracy for early detection of mental deterioration.

## Figures and Tables

**Figure 1 diagnostics-13-00167-f001:**
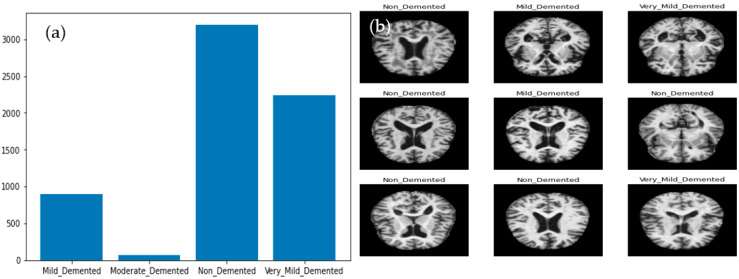
(**a**) Distribution of 4 different classes and (**b**) a sample image in the selected dataset.

**Figure 2 diagnostics-13-00167-f002:**
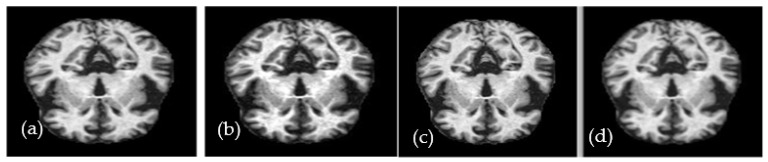
A visual example of the proposed preprocessing step to enhance the contrast of MRI images, where figure (**a**,**c**) are the original and (**b**,**d**) are with noise removed and enhanced.

**Figure 3 diagnostics-13-00167-f003:**
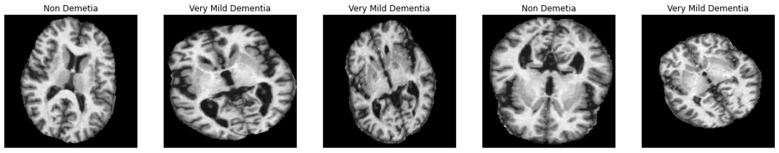
A visual example of the data augmentation technique applied to MRI image.

**Figure 4 diagnostics-13-00167-f004:**
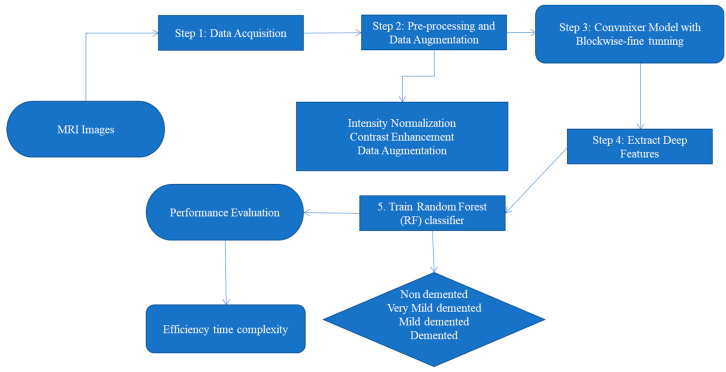
A systematic flow diagram of the proposed CAD-ALZ model to recognize different stages of AD disease during diagnosis through MRI images.

**Figure 5 diagnostics-13-00167-f005:**
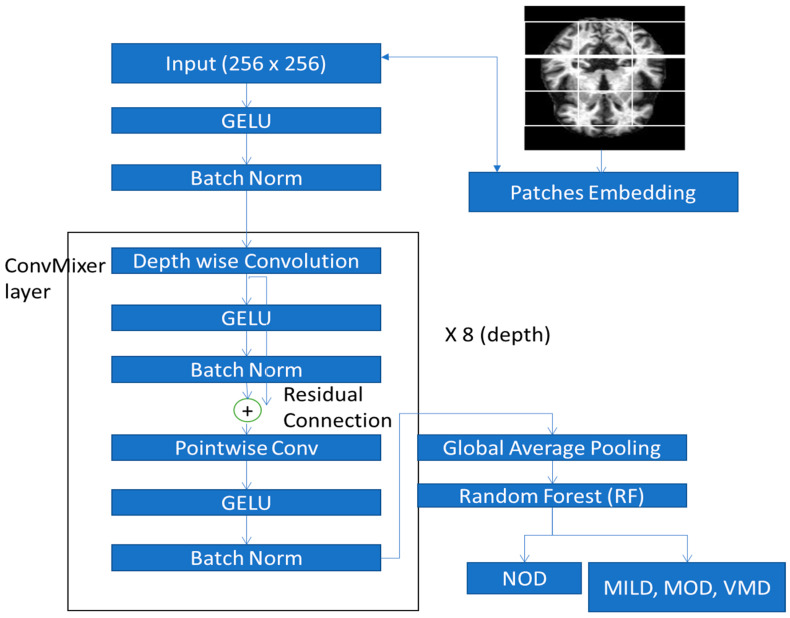
The proposed ConvMixer model architecture to integrate blockwise fine-tune strategy.

**Figure 6 diagnostics-13-00167-f006:**
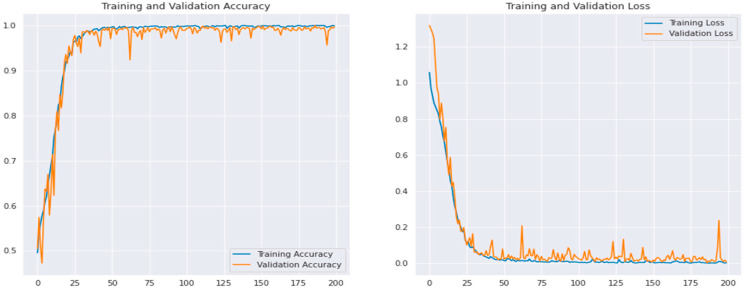
Training and validation loss of proposed CAD-ALZ system on the selected datasets.

**Figure 7 diagnostics-13-00167-f007:**
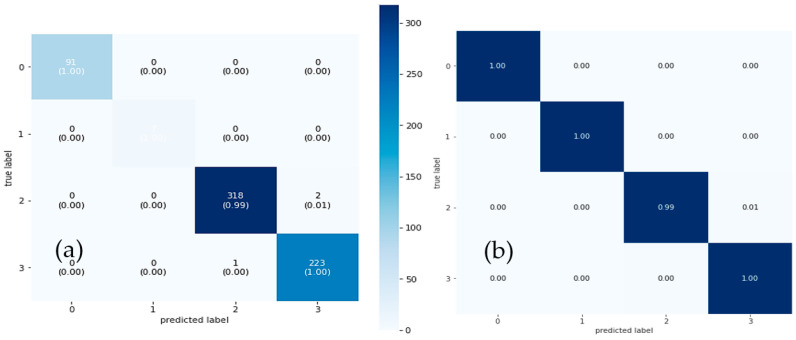
An example of confusion matrix of proposed CAD-ALZ system to recognize all stages of AD (**a**) without preprocessing step and (**b**) with a preprocessing technique, where 0 shows the normal, 1 means mild dementia, 2 represents moderate dementia, and 3 shows very mild dementia stages of AD.

**Figure 8 diagnostics-13-00167-f008:**
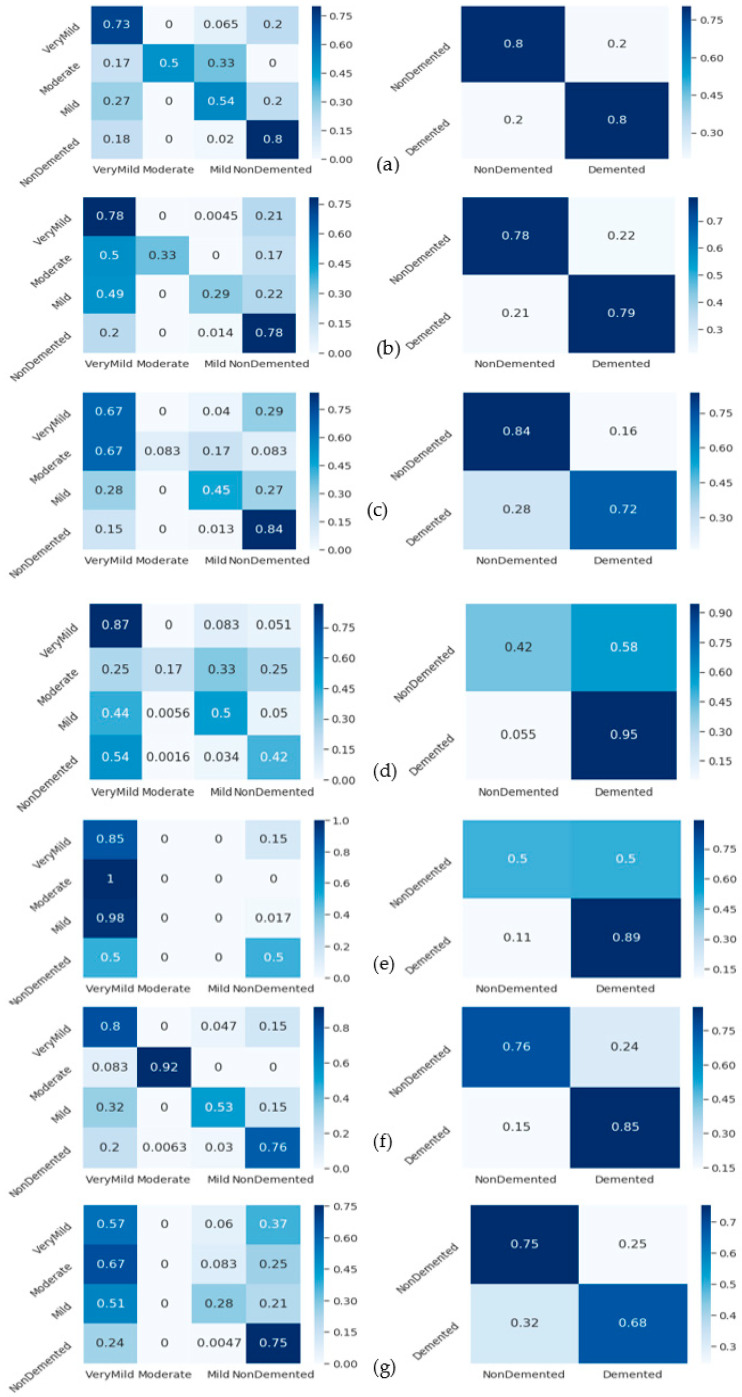
An example of confusion matrix for state-of-the-art systems to recognize four stages of AD such as non-demented, mild dementia, moderate dementia, and very mild dementia. Where (**a**) shows Murugan-DEMNET-CNN [15], (**b**) represents Mehmood-Siamese-CNN [16], (**c**) presents Islam-Deep-CNN [17], (**d**) shows Janghel-VGG16-SVM [19], (**e**) represents Talo-ResNet-50 [21], (**f**) shows Gunawardena-CNN-SVM [23], and (**g**) shows Wang-CNN-Maxpool [26] systems.

**Figure 9 diagnostics-13-00167-f009:**
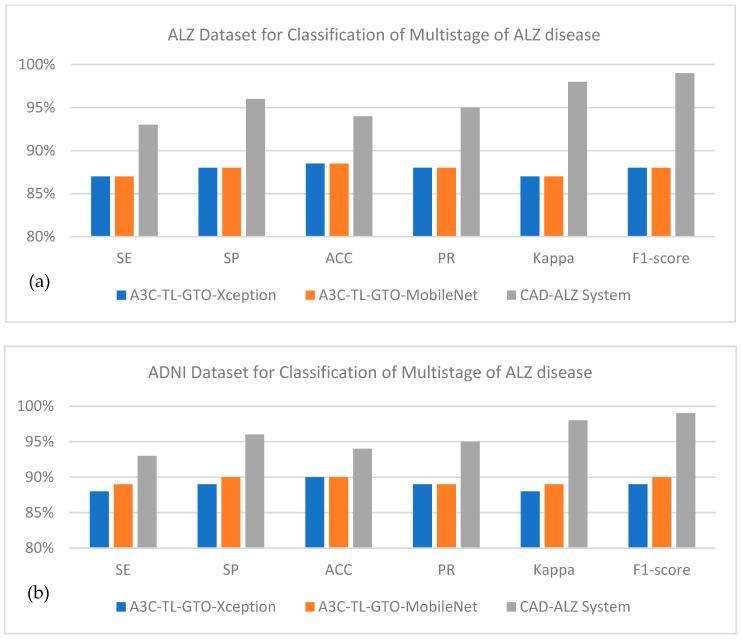
Comparisons by using (**a**) ALZ and (**b**) ADNI datasets by using state-of-the-art systems to recognize four stages of AD, such as non-demented, mild dementia, moderate dementia, and very mild dementia.

**Figure 10 diagnostics-13-00167-f010:**
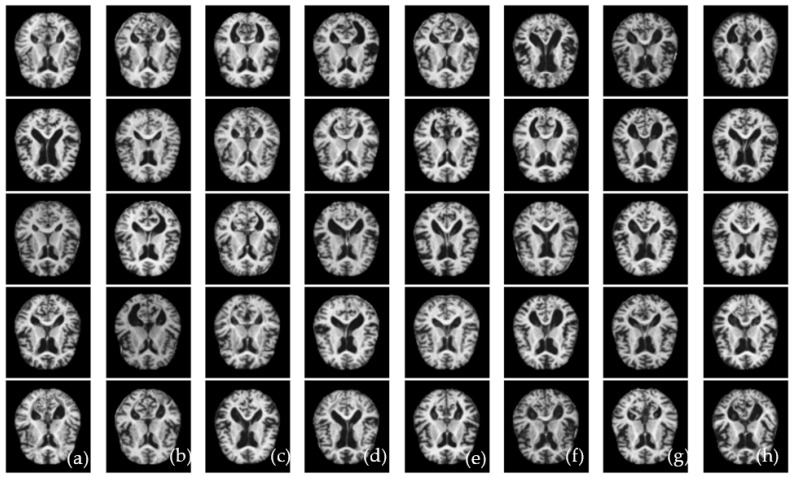
A visual example of classification of CAD-ALZ system for recognition of AD disease on Kaggle-ALZ (**a**–**h**), where figure shows normal (**a**,**e**), mild dementia (**b**,**f**), moderate dementia (**c**,**g**), and very mild dementia (**d**,**h**).

**Table 1 diagnostics-13-00167-t001:** A Comparative performance of existing work for the classification on ALZ disease.

Reference	* Dataset	Methodology	Results	Drawbacks
Liu et al. [30]	ADNI	Weakly supervised densely connected neural network (wiseDNN)	ACC: 93%	Evaluated on single dataset and classify two-classes
Long et al. [31]	ADNI	Deformation-based technique	ACC: 89.5%	Evaluated on single dataset and classify two-classes
Duraisamy et al. [32]	ADNI	FCM-based weighted probabilistic neural network (FWPNN)	ACC: 98.63%	Too much image processing, three classes and evaluate on signal dataset
Wu et al. [33]	ADNI	GoogleNet and CaffeNet	ACC: 95.42% to 97.01%	TL with limited dataset on two classes, detection accuracy is limited; computationally expensive
Zhang et al. [34]	ADNI	Image Processing and Machine learning	ACC: 92%	Image processing, handcrafted-based feature extraction approach, which limits the detection accuracy
Lin et al. [35]	ADNI	CNN	ACC: 91%	Two classes, CNN, used one dataset, and limited applicability
Goceri et al. [36]	ADNI	Sobolev-gradient-based optimization	ACC: 98%	Two classes, complicated approach compared to prior techniques
Kumar et al. [37]	ADNI	CNN and SVM	ACC: 95%	Two classes, computationally expensive and limited applicability
Oh et al. [38]	ADNI	Image features and CNN	ACC: 86.60%	Two classes, complex image processing
Altaf et al. [39]	ADNI	Gray level co-occurrence matrix (GLCM), scale-invariant feature transform, local binary pattern, and gradient histogram	ACC: 90%	Two classes, hand-crafted features limit the detection accuracy, and single dataset
Shi et al. [40]	ADNI	Stacked denoising sparse auto-encoder (DSAE) with feature fusion	--	Two classes, feature fusion approach to make complexity expensive, and limited applicability
Liu et al. [41]	ADNI	multi-task CNN and DenseNet models	ACC: 89%	Three classes only and used only one dataset
Tanveer et al. [46]	ADNI	TL: VGG-16	ACC: 95.73%	Three classes only and used only one limited dataset, classifier is not generalized
Chen et al. [48]	ADNI	Iterative sparse and deep learning (ISDL)	ACC: 94%	Two classes only and used only one limited dataset, classifier is not generalized
Bron et al. [49]	ADNI	SVM and CNN method	ACC: 92%	Two classes only and used only one limited dataset, classifier is not generalized
Abuhmed et al. [52]	ADNI	BiLSTM model	--	Three classes only and used only one limited dataset, classifier is not generalized
Hazarika et al. [53]	ADNI	DNN with LeNet to replace max pooling layers	ACC: 93.5%	Two classes only and used only one limited dataset, classifier is not generalized
Herzoz et al. [55]	ADNI	CNN and SVM	ACC: 93.0%	Two classes only and used only one limited dataset, classifier is not generalized
An et al. [56]	ADNI	DBN+ NN	--	Three classes only and used only one limited dataset, classifier is not generalized
Lebedev et al. [59]	ADNI			
Baghdadi et al. [61]	ADNI	multi-task CNN and DenseNet models	ACC: 89%	Three classes only and used only one dataset.

* ADNI: Alzheimer’s Disease Neuroimaging Initiative dataset, TL: transfer learning, CNN: Convolutional neural network, SVM: Support vector machine, DNN: Deep neural network, DBN: Deep belief network, BiLSTM: Bidirectional long short-term memory, LeNet: LeCun network.

**Table 2 diagnostics-13-00167-t002:** Utilized data augmentation techniques to make data balanced.

Methods	Values Assigned
Affine transform	True
Pan	True
Spin-range	0.12
Crop	True
Horizontal-flip	True
Vertical-flip	False
Affine transform	True

**Table 3 diagnostics-13-00167-t003:** Parametric configuration of the convolutional layer of proposed model.

Convolutional Layers	Parameters
Convolution-32	(3 × 3 × 3 + 1) × 32
Convolution-32	(3 × 3 × 32 + 1) × 64
Convolution-64	(3 × 3 × 64) + (1 × 1 × 64 + 1) × 128
Convolution-64	(3 × 3 × 128) + (1 × 1 × 128 + 1) × 128
Convolution-128	(3 × 3 × 128) + (1 × 1 × 128 + 1) × 256
Convolution-128	(3 × 3 × 256) + (1 × 1 × 256 + 1) × 256
Convolution-256	(3 × 3 × 256) + (1 × 1 × 256 + 1) × 728
Convolution-256	(3 × 3 × 728) + (1 × 1 × 728 + 1) × 728
**Total**	**87,488**

**Table 4 diagnostics-13-00167-t004:** Performance evaluation metrics of the implemented CAD-ALZ system on 51,200 samples.

AD Type	^1^ SE	^2^ SP	^3^ ACC	^4^ PR	Kappa	F1-Score
Normal	93%	96%	94%	0.95	0.98	99%
Dementia	95%	96%	95%	0.96	0.97	100%
**Average Result**	94%	96%	95%	0.96	0.96	99.5%

^1^ SE: Sensitivity, ^2^ SP: Specificity, ^3^ ACC: Accuracy, ^4^ PR: Precision.

**Table 5 diagnostics-13-00167-t005:** Assessment of the result of the developed CAD-ALZ system.

Methods	^1^ SE	^2^ SP	^3^ ACC	^4^ PR	Kappa	F1-Score
Normal	93%	96%	94%	0.95	0.98	99%
Mild Dementia	95%	96%	95%	0.96	0.97	100%
Moderate Dementia	93%	96%	94%	0.95	0.98	99%
Very Mild Dementia	95%	96%	95%	0.96	0.97	100%
**Developed CAD-ALZ System**	93%	96%	94%	0.95	0.98	99%

^1^ SE: Sensitivity, ^2^ SP: Specificity, ^3^ ACC: Accuracy, and ^4^ PR: precision.

**Table 6 diagnostics-13-00167-t006:** Analysis results obtained using the cross-validation method of the proposed approach (k = 10).

Feature Set	Methods	^1^ SE	^2^ SP	^3^ ACC	^4^ PR	^4^ Kappa	^4^ F1-Score
Kaggle-ALZ (4000)	Normal	93%	96%	94%	0.95	0.98	99%
	Mild Dementia	95%	96%	95%	0.96	0.97	100%
	Moderate Dementia	93%	96%	94%	0.95	0.98	99%
	Very Mild Dementia	95%	96%	95%	0.96	0.97	100%
ADNI (18,000)	Normal	93%	96%	94%	0.95	0.98	99%
	Mild Dementia	95%	96%	95%	0.96	0.97	100%
	Moderate Dementia	93%	96%	94%	0.95	0.98	99%
	Very Mild Dementia	95%	96%	95%	0.96	0.97	100%
Combined datasets	Normal	93%	96%	94%	0.95	0.98	99%
	Mild Dementia	95%	96%	95%	0.96	0.97	100%
	Moderate Dementia	93%	96%	94%	0.95	0.98	99%
	Very Mild Dementia	95%	96%	95%	0.96	0.97	100%
	Normal	93%	96%	94%	0.95	0.98	99%

^1^ SE: Sensitivity, ^2^ SP: Specificity, ^3^ ACC: Accuracy, and ^4^ PR: precision.

**Table 7 diagnostics-13-00167-t007:** Computational performance of the different DL models.

Architectures	Complexity (FLOPs)	# Parameters (M)	Model Size (MB)	GPU Speed (MS)
ConvMixer-V8	**67.3 M**	**1.9**	**9.3**	**0.6**
ConvMixer	98.9 M	2.5	14.5	1.7
SqueezeNet	94.4 M	2.4	12.3	1.2
ResNet18	275.8 M	2.7	15.2	2.6
MobileNet	285.8 M	3.4	16.3	2.7
Inception V3	654.3 M	3.9	17.5	2.9
Xception	66.9 M	2.5	14.5	2.7
AlexNet	295.8 M	2.5	12.3	3.3

FLOPS: Floating-point operations, M: Millions, MB: Mega Byte, MS: milliseconds, #: number of.

## Data Availability

The Alzheimer dataset is available at https://www.kaggle.com/datasets/tourist55/alzheimers-dataset-4-class-of-images (accessed on 2 January 2021). In addition, the CAD-ALZ model’s source code is freely available on the GitHub page at: https://github.com/Qaisar256/CAD-ALZ-RF (accessed on 15 November 2022).

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
