# Peer review of "CAD-ALZ: A Blockwise Fine-Tuning Strategy on Convolutional Model and Random Forest Classifier for Recognition of Multistage Alzheimer’s Disease"

_diagnostics, 2023, doi:10.3390/diagnostics13010167_

Round 1

Reviewer 1 Report

Journal: Diagnostics (ISSN 2075-4418)
Manuscript Title: A Multistage Classification of Alzheimer Disease by Blockwise Finetune Strategy on ConvMixer Model
Manuscript ID: diagnostics-2081504
Submission Date: Thursday, December 1, 2022
The manuscript presented “a multistage classification approach of alzheimer disease”. However, the major and critical weak points are that:
(1)    Their proposed work discussion is weak distributed to be described or analyzed.
(2)    The novelty is not guaranteed.
(3)    Their work is not compared with state-of-the-art approaches nor related studies.
(4)    Their experiments leak from the descriptive and statistical analysis.
The rest of my review presents other weak points, comments, and opinions in detail.
Overall Comments:
(1)    [KEYWORDS] The keywords (i.e., index terms) should be sorted in alphabetical order.
(2)    [ABSTRACT] The abstract should contain the best-achieved results from the performed experiments.
(3)    [ABSTRACT] The abstract should reflect the contributions of the manuscript. I suggest rewriting it.
(4)    The link in the abstract does not work. The authors’ repo is set as private.
(5)    [RELATED WORK] Where are the related studies? They should be declared in a separate section.
(6)    [RELATED WORK] A table of comparisons should be added at the end of the related studies section to praise the pros. and cons. of them. The year column should be added and they should be ordered by it.
(7)    [EQUATIONS] The authors should follow the journal authors’ guidance in writing the equations, symbols, and variables. Please, refer to the authors guidelines on the journal official website.
(8)    Introducce Section 2. Don’t leave it empty. In other words, at the beginning of Section 2, a summary of the section should be written.
(9)    Algorithm 1 should be written in the pseudocode format. The same issue with Algorithm 2.
(10)    [DATASETS] Samples from the used dataset should be added and annotated.
(11)    [METHODOLOGY] The suggested approach is not clearly discussed. More scientific details should be added.
(12)    In Figure 7(a): a cell value is not clear.
(13)    [METHODOLOGY] What are the used equations in the suggested approach? In other words, how the suggested approach is derived?
(14)    [METHODOLOGY] Where is the overall pseudocode? Flowchart? of the suggested approach?
(15)    [EXPERIMENTS] The working environment (i.e., software and hardware) should be declared and added to a table.
(16)    [EXPERIMENTS] The experimental configurations (i.e., settings) should be declared and added to a table.
(17)    [EXPERIMENTS] What are the criteria for selecting the experimental configurations?
(18)    [EXPERIMENTS] More experiments should be conducted using different configurations.
(19)    [EXPERIMENTS] Where is the detailed and statistical discussion of the reported results?
(20)    [EXPERIMENTS] More experiments should be conducted using a different dataset to prove the generalization.
(21)    [ABBREVIATIONS] The authors should add a table of abbreviations in the revised manuscript.
(22)    [SYMBOLS] The authors should add a table of symbols in the revised manuscript.
(23)    [CONCLUSIONS] The conclusions in this manuscript are primitive. Please, write your conclusions.
(24)    [REFERENCES] There are no citations for many sentences in the manuscript. Why? Please check.
(25)    [REFERENCES] The references should be written in the same style following the journal authors’ guidance.
(26)    [REFERENCES] Recent citations from 2021 and 2022 should be added to the manuscript.
(27)    [PROOFING] The authors should get editing help from someone with full professional proficiency in English.
(28)    [PROOFING] The manuscript should be checked again to fix any typos such as missing spaces and commas.
(29)    [CONSISTENCY] The manuscript structure is too short. It must be elaborated in their applied technology as should support more rigorous technical aspects.
(30)    [NOVELTY] What is the novelty of the suggested approach?
(31)    [FIGURES] The authors should provide high-resolution figures in the manuscript. For example, Figure 9.
(32)    [LIMITATIONS] What are the limitations of the current study? It should be added in a separate section.
(33)    Compare your approach with the approach disccused in “Baghdadi, N. A., Malki, A., Balaha, H. M., Badawy, M., & Elhosseini, M. (2022). A3C-TL-GTO: Alzheimer Automatic Accurate Classification Using Transfer Learning and Artificial Gorilla Troops Optimizer. Sensors, 22(11), 4250.”.
For the authors in case of the authors got a chance to review the manuscript and submit the revised one after the editor’s decision, please, provide a table in the revised manuscript mentioning (1) the comment, (2) the authors’ response, and (3) the authors’ change (if applicable). Please, consider all of the comments and don’t ignore any of them.
Please, refer to the attached file "diagnostics-2081504 Reviewer.pdf" for the same comments in an organized format.

Author Response

Dear Reviewer,

We have updated the paper according to all of your comments. Attached to this message.

Reviewer 2 Report

This paper introduced a CAD-ALZ model via extracting deep features through ConvMixer layer with blockwise finetune strategy on a very small original dataset for recognizing different ALZ stages. The best accuracy is 99.21%, which is better than other approaches. This proposed CAD-ALZ system has obtained high accuracy for early detection of mental deterioration. Overall, the study is interesting and useful, and the manuscript is easy to follow. Here are some small concerns:

1. What is the difference between the subfigures in Figure 2? It maybe difficult to recognize the difference between them just using eyes. Some labels and explanation can help readers understand.

2. In line 165, did the authors means Figure 3 or Figure 2? Why Figure 3 appear earlier?

3. The authors mentioned that “The overfitting problem is avoided by our model with the use of data augmentation techniques”. I am wondering can the overfitting be fully avoided by just adding the data augmentation. How to evaluate the overfitting or how to prove that it has been totally avoided.

4. The proposed method might be sensitive to the values of its main controlling parameters. How did you determine the parameters. Also why choose N = 8 times?

5. How scalable is this approach?

6. What is the computational cost of this approach, is it computationally efficient?

7. Any limitations of this work in practical applications?

8. In line 614, where is Figure 15? Maybe the number is incorrect.

Author Response

(The authors gave the same response as above.)

Round 2

Reviewer 1 Report

The authors had made a major revision. Thanks to them. However, there are some minor concerns on the updated version:

(1) Refer to the authors' names in the citations. For example, "In [47]" should be "xxx et al. [47]".

(2) Define Ke and Ko.

(3) Provide a clean and updated manuscript version because figures and sentences are shifted.

(4) Try to add the table of abbreviations for the readers to refer to the meanings easily.

Author Response

(1) Refer to the authors' names in the citations. For example, "In [47]" should be "xxx et al. [47]".
Response 1: Dear reviewer, we have updated the paper according to your mentioned style in all those 
sentences, which started by “In …”. You can easily see those changes in the revised version.
Thank you for this valuable comment.
(2) Define Ke and Ko.
Response: Yes, dear reviewer, we have defined these two parameters in the statistical analysis part. 
Thank you to clear this point.
(3) Provide a clean and updated manuscript version because figures and sentences are shifted.
Yes, we have provided pdf clean file.
(4) Try to add the table of abbreviations for the readers to refer to the meanings easily.
Yes, we have added abbreviation table, which are mostly used in the paper before references section. 
Thank you very much
